# Using Polygenic Risk Scores Related to Complex Traits to Predict Production Performance in Cross-Breeding of Yeast

**DOI:** 10.3390/jof8090914

**Published:** 2022-08-29

**Authors:** Yi Dai, Guohui Shi, Mengmeng Chen, Guotao Chen, Qi Wu

**Affiliations:** 1State Key Laboratory of Mycology, Institute of Microbiology, Chinese Academy of Sciences, Beijing 100101, China; 2University of the Chinese Academy of Sciences, Beijing 100049, China

**Keywords:** polygenic risk score, hybrid, cross-breeding, complex trait, GWAS

## Abstract

The cultivation of hybrids with favorable complex traits is one of the important goals for animal, plant, and microbial breeding practices. A method that can closely predict the production performance of hybrids is of great significance for research and practice. In our study, polygenic risk scores (PRSs) were introduced to estimate the production performance of *Saccharomyces cerevisiae*. The genetic variation of 971 published isolates and their growth ratios under 35 medium conditions were analyzed by genome-wide association analysis, and the precise *p*-value threshold for each phenotype was calculated. Risk markers for the above 35 phenotypes were obtained. By estimating the genotype of F1 hybrids according to that of the parents, the PRS of 613 F1 hybrids was predicted. There was a significant linear correlation between the maximum growth rate at 40 °C and PRS in F1 hybrids and their parents (R^2^ = 0.2582, R^2^ = 0.2414, respectively), which indicates that PRS can be used to estimate the production performance of individuals and their hybrids. Our method can provide a reference for strain selection and F1 prediction in cross-breeding yeasts, reduce workload, and improve work efficiency.

## 1. Introduction

Cross-breeding is one of the effective ways to obtain new yeast strains with superior traits. The production of excellent hybrids through cross-breeding has led to a continuous and substantial increase in global agricultural productivity [1,2,3]. It is usually limited to complex traits and can be influenced by parental background and imprinting [4]. It facilitates the construction of novel yeast (*Saccharomyces cerevisiae*) strains with preferred characteristics from multiple parental strains by sexual hybridization. Cross-breeding of industrial strains, such as baker’s [5], sake [6], and wine yeast strains [7], has been reported. The selection of parents with traits of interest is a prerequisite for obtaining superior hybrid offspring. However, there is no reliable forecasting method for progeny to guide practice in fungal cross-breeding. Many industrial strains have suffered from low sporulation efficiency and spore viability [8]. Therefore, it is very meaningful to predict the production performance of F1 hybrids without extra experiments and time.

Many complex diseases in human beings are caused by both genetic and environmental factors. Moreover, most of them are affected by multiple genes [9], so theories of quantitative genetics for complex traits can be used to study such diseases [10,11]. In the era of omics, high-throughput techniques such as genome-wide genetic association studies (GWAS) have been widely used for the comprehensive assessment of genetic susceptibility for various complex traits [12,13]. In recent years, the polygenic risk score (PRS), based on GWAS summary results that provide a comprehensive assessment of genetic predisposition for complex traits, such as height, body weight, cardiovascular disease, and rheumatoid arthritis [14,15], at the individual level has been widely used in the biomedical field. It can effectively identify groups of individuals with substantially increasing risks so that certain medical treatments or behavioral modifications can be recommended as a precaution [14,15,16,17]. It weighted the significant risk markers of GWAS to evaluate the genetic liability of individuals to complex traits [18]. That is, PRS is defined as a combination of single nucleotide polymorphisms (SNPs) that associate with the trait of interest [19]. From a statistical point of view, PRS can be considered as a single marker similar to an individual biomarker (or biomarker score) and has been commonly used for clinically relevant disease prediction [20].

However, PRS is only used to identify the individuals with clinically significant increased risks, primarily for the early warning of important human traits for adjunctive treatment [15]. In the biomedical field, PRS calculation has become a popular approach for using GWAS datasets. Even though this approach has not been widely used in non-human organisms, it has been proved as an effective predictor of individuals’ genetic liability to have complex traits. In the practice of fungal hybridization, we also pay attention to some complex traits which have important contributions for production, so whether PRS can be used as an effective indicator of fungal breeding to select high performance yeast strains needs to be investigated.

The goal of this study is to identify risk markers related to the production performance of yeasts, increase the variance explained by phenotypic diversity, and achieve the accurate prediction of phenotypes in F1 hybrids. In this paper, it is proposed to introduce PRS into breeding research and to screen out high-potential parents based on the parent’s trait performance. In order to predict the phenotype of the next generation of hybrids, we estimate the offspring genotype according to parental genotype and then estimate its PRS.

## 2. Materials and Methods

We developed a method for predicting the growth ratio of hybrid F1 and validated it with 52 wild-type homozygous *S. cerevisiae* genomes and the growth ratio of their F1 hybrids without any genetic marker through spore-to-spore mating. The method consisted of three main steps. First, to identify risk markers associated with the growth ratio, we downloaded the published GWAS summary results regarding the association between 83,794 variant markers and growth ratios under 35 medium conditions. We screened out the risk markers by calculating the precise *p*-value. Then, by estimating the genotypes of F1 hybrids according to the parental genotypes, we calculated their PRS. Finally, the potential of F1 was judged according to the value of PRS. Figure 1 summarizes the main steps of this process. The Methods section describes the pipeline in detail.

### 2.1. Training Datasets and Testing Datasets

To calculate the PRS of one individual, we needed to obtain the risk loci associated with each trait. We used a dataset of 1011 *S. cerevisiae* isolates to obtain phenotype association loci. The isolates included in this project were carefully selected from providers and references including 23 kinds of ecological and 312 different geographical origins from around the world [21]. Ecology includes the human-related environment as well as the natural environment. The geographic origin was also highly diverse, with a global distribution. These global samples were suitable for our search for phenotype association loci. Here, we primarily used summary statistics from recent GWAS studies conducted on 971 isolates in the training datasets, as well as the phenotypic file which included 35 phenotypes (Appendix A). We downloaded BED, BIM, and FAM files with all biallelic positions known for the 971 isolates (1011GWASMatrix.tar.gz) as well the phenotypic file (phenoMatrix_35ConditionsNormalizedByYPD.tab.gz) from http://1002genomes.u-strasbg.fr/files/ accessed at 19 December 2017and 30 March 2017 respectively.

The testing dataset with variant calls of 266 *S. cerevisiae* isolates was a gift from Prof. Fengyan Bai, Institute of Microbiology, Chinese Academy of Sciences. Their work provided the detailed information about the SNPs of these isolates [22]. They were selected from different wild-type lineages which were shown to be homozygous by genome analysis and to have the greatest genetic diversity in the wild population of *S. cerevisiae* that has been documented to date [22]. The genome sequence of *S. cerevisiae* S288c was used as the reference genome (version R64-1-1) and was downloaded from the National Center for Biotechnology Information (NCBI). The phenotypic data (maximum growth rate at 40 °C, YPD40) were measured for 613 F1 hybrids and their parents without any genetic marker by spore-to-spore mating between pairs of the 52 wild *S. cerevisiae* strains gifted from Liang S [23].

### 2.2. Identify Risk Markers

Since the phenotypic data we obtained were already normalized, and the genomic variants had also been filtered, we could perform the genome-wide association analysis (GWAS) directly. We subjected 971 isolates with MAF > 5% to the performed genome-wide association analysis by *GEMMA* 0.98.3 with a linear mixed model and *p*-values from the Wald test to account for the 35 phenotypes. Then, we computed the variance explained by our significantly associated markers [24] by an in-house Python script. LD score regression was used to distinguish swelling from true polygenic signals and biases [25]. The method was used after GWAS to quantify the contribution of each factor by examining the relationship between test statistics and linkage disequilibrium (LD). LD score regression intercept was used to estimate the polygenicity of traits. *PRSice-2* [26] was used to determine the polygenic risk scores (PRS) under different *p*-value thresholds according to the results of GWAS and provide the best-fit PRS and *p*-value significance threshold. Then, we used it to calculate the PRS of each isolate and obtained the precise *p*-value threshold (PT) in order to explain a higher variance. Markers with a *p*-value below PT were screened to form a dataset with risk loci associated with the phenotype by in-house Python scripts.

### 2.3. Calculation of the Polygenic Risk Score

PRSs were generated by multiplying the genotype dosage of each risk loci by its respective weight and then summing across all variants in the score using in-house Python scripts.
$$\mathrm{PRS}=\sum_{i}^{m}\beta_{i}\left(\sum_{j=0}^{2}w_{ij}\times j\right)$$
where $w_{ij}$ is the probability of observing genotype j where j∈{0,1,2} for the ith SNP; m is the number of SNPs; and $\;\beta_{i}\;$ is the effect size of the ith SNP estimated from the relevant GWAS data. The PRS of the candidate parent was calculated based on the distribution of the SNPs of the candidate parent. SNPs were treated as 0/1/2 according to genotype.

### 2.4. Estimate the Genotype of F1 Hybrids

For homozygous SNP, there is ideally only one allele that is passed down to the next generation. However, for heterozygous loci, we cannot accurately determine which allele is passed on to the offspring. Assuming that A and a are two genotypes at one locus, and A is the mutant allele, we used 0/1/2 to stably encode aa, Aa, and AA, respectively. In this way, we calculated the mean value of parents at this locus as the estimated genotype of F1. Thus, if both parents were heterozygous at that site, then the offspring were still heterozygous at that site. According to Mendel’s first law of segregation, the probability that the next generation remains heterozygous in the absence of recombination is the highest (50%). However, for one parent with a homozygous genotype and one with a heterozygous genotype, our method produced values of 0.5 and 1.5, which did not exist in the genotype, just for convenience of calculation. Here, we adopted a compromise method, ignoring the possibility of heterozygous site hybridization, to produce homozygous results, so there is the possibility of underestimation and overestimation.

### 2.5. Judgment of Heterosis of F1 Hybrids

To measure the degree of heterosis, F1 was divided into groups according to the above three parameters, and the PRS difference between groups was compared. MPH (mid-parent heterosis) > 0 was considered to show heterosis, BPH (best-parent heterosis) > 0 was considered to show significant heterosis, and DEP (depression) > 0 was considered to show decline. The formula is as follows:BPH=F1−BP
MPH=F1−MPV
DEP=WP−F1
where F1 is the F1 hybrid phenotype, BP is the maximum of the parental phenotype, MPV is the average of the parental phenotype, and WP is the minimum of the parental phenotype.

### 2.6. Statistical Analysis

The normality was verified by Kolmogorov–Smirnov test. We performed linear fitting on the PRS and phenotype data of 52 candidate parents and 613 F1 hybrids and output at a 95% confidence interval. All tests were two tailed with an alpha threshold of 0.05. Statistical analyses were conducted in R v3.6.0.

### 2.7. Code Availability

The scripts for calculating PRS for parents, gametes, and any F1 hybrids were deposited in GitHub at https://github.com/DYqwert/pyprs/ accessed on 1 July 2022.

## 3. Results

### 3.1. Identify Risk Markers for Production Performance in 35 Medium Conditions

We performed GWAS analysis on the growth ratios under 35 medium conditions (Appendix A). We mainly used two phenotypes of growth ratio at 40 °C in our testing set to verify the relationship between PRS and the phenotype of the isolates (Figure 2). With *p* < 5 × 10^−8^ as the high-confidence significant threshold, we found 12 SNPs associated with YPD40 (R^2^ = 0.33) (Figure 2A,B). When we repeated the analysis using high-resolution PRS, we found the most predictive PRS at PT = 0.37885, and 2126 SNPs were shown (R^2^ = 63.97%) (Figure 2C,D). According to the LD score regression results, we also found that the LD score regression intercept was close to 1 (0.817 ± 0.0256), indicating that the phenotype was affected by polygenicity.

### 3.2. Relationship between PRSs and Phenotypes

There was a significant linear regression between PRS and YPD40 for 52 strains (R^2^ = 0.2582, *p*-value = 1.203 × 10^−4^), indicating that there was a good linear correlation between PRS and phenotype (Figure 3A), which further confirmed that calculating PRS can be used to predict the growth rate of strains at 40 °C. At each locus, the combination of parental genotypes and fixed parental loci were analyzed, and they could only produce one kind of gamete. The situations (AA×AA, AA×aa, aa×aa) accounted for 99.96% ± 0.04% (Figure 3B). When both parents were homozygous genotypes, the genotype of the F1 hybrids at this locus could be predicted with relative accuracy. At the same time, we calculated the estimated PRS of F1 by estimating the genotype of F1 hybrids, which also had a good linear relationship and was close to the goodness of fit of the parents (Figure 3C).

### 3.3. PRS for Heterosis

We found that there was no significant difference in phenotypic distribution among our 52 wild candidate strains compared with other strains, but most of the F1 hybrid strains were significantly higher than their parents (Figure 4A), indicating significant heterosis. In terms of PRS, the distribution of candidate strains and F1 hybrid strains were relatively concentrated, and there was no distribution on the two tails of the other strains (Figure 4B). There was a significant difference in PRS between the strains that showed mid-parent heterosis and those that did not (W = 10,622, *p*-value = 0.002591), and there was also a significant difference between the PRS of the strains that showed depression and the other strains (W = 8150, *p*-value = 1.397 × 10^−6^). However, there was no significant difference in PRS between the strain with best-parent heterosis and the other strains (W = 31,429, *p*-value = 0.09518) (Figure 4C). Our results suggest that heterosis and depression can be determined to a certain extent.

## 4. Discussion

Achieving trait prediction of F1 hybrids is of great significance for cross-breeding, whether in terms of industrial applications, experimental costs, or genetic research. Our method enables the efficient prediction of traits in the progeny of hybrids, and we confirmed the correlations with the experimental results. This method can accurately predict the phenotype of hybrid generation with only the parental phenotype and genome and without hybridization experiments being conducted. Moreover, this method is not only applicable to the production traits mentioned in this paper, but also to other important complex traits (Appendix A). It is hoped that this method can be helpful for the cross-breeding of fungi.

PRSs are similar to narrow heritability in that they represent the aggregate and additive effects of segregating loci with small effect to some degree [27,28]. Considered as a quantification of the additive effect of an individual’s risk genes, PRS can be used to estimate the additive genetic variance of a hybrid offspring. In the process of gene transmission, the additive genetic effects are relatively stable. Therefore, it is feasible and meaningful to estimate the productivity of hybrids by evaluating the potential of their parents based on the multiple variants related to productivity. While prediction of phenotype from an individual’s genetic profile is compromised by this polygenicity, the application of PRS has shown that the prediction is sufficiently accurate for several applications [17,18]. Our method, unlike genomic selection, refers to the use of dense markers covering the whole genome to estimate the breeding value of selection candidates for a quantitative trait [29,30]. With the risk loci selected by the high-resolution PRS, the method can reduce the false negatives on GWAS results and the false positives on genomic selection results and, furthermore, significantly increase the PVE and provide a significant improvement in screening efficiency without progeny testing. Unlike previous applications in precision medicine and preventive medicine [15], we can use PRS in our method in breeding research to extend its application to non-human organism prediction.

For homozygous SNP, ideally only one allele is inherited by the next generation. However, for heterozygous loci, we cannot accurately determine which allele is passed on to the offspring. Our method may underestimate or overestimate the effect of homozygous loci. Simply put, under the assumption that there are only two alleles, the offspring can only have three genotypes. Considering the amount of calculation, we only need to estimate the minimum and maximum PRS of different F1 genotypes, and the obtained interval can be used as a breeding reference. Theoretically, the variance of PRS mainly comes from the genotype difference produced by uncertain gametes, so, with the increase of uncertain genotypes, the variance of the estimated PRS will increase. In addition, our method cannot yet predict the heterosis of hybrids. However, our results show that PRS may be used to judge whether heterosis exists, although it cannot be used to judge the degree of heterosis. Heterosis is an important reason for cross-breeding research, so the prediction of F1 heterosis is also crucial. However, the complementation of the allelic variation and the variation in gene content and gene expression patterns are likely to be an important contributor to heterosis [4]. There is some evidence that has shown the possible roles of non-additive genes in the manifestation of heterosis or the outbreeding of depression in *S. cerevisiae* [23]. Although the additive effect and the epistatic effect have a certain mutual influence [31], it is difficult to estimate the phenotype difference between hybrids and their parents caused by the non-additive effects of the PRS of hybrids. Generally, the additive effect is relatively stable. We were able to predict the narrow-sense heritability efficiently, but the prediction for broad-sense heritability alone was unreliable with PRS. So, accurately predicting broad-sense heritability will be the focus of our further research.

## 5. Conclusions

The prediction of complex traits in hybrids is of great significance for production cost, experimental cost, and genetic research relating to breeding. This method can effectively predict the phenotypes of F1 hybrids, and we confirmed the correlation with experimental results, which may be helpful for the cross-breeding of fungi.

## Figures and Tables

**Figure 1 jof-08-00914-f001:**
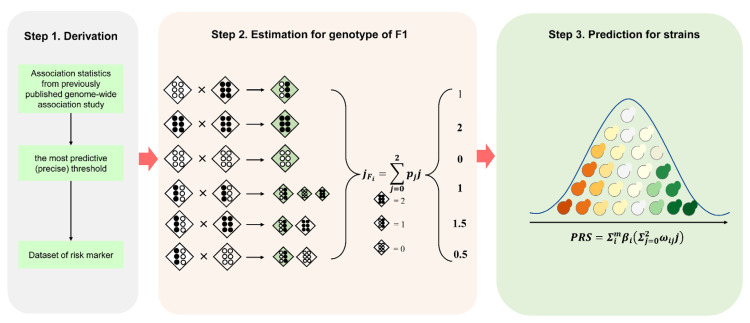
Study design and workflow. We developed a method for predicting the growth ratio of hybrid F1. The method consists of three main steps. First, to identify risk markers associated with the growth ratio, we downloaded the published GWAS summary results. We screened out the risk markers by calculating the precise *p*-value. Then, by estimating the genotypes of F1 hybrids according to the parental genotypes, we calculated their PRS. Finally, the potential of F1 was judged according to the value of PRS.

**Figure 2 jof-08-00914-f002:**
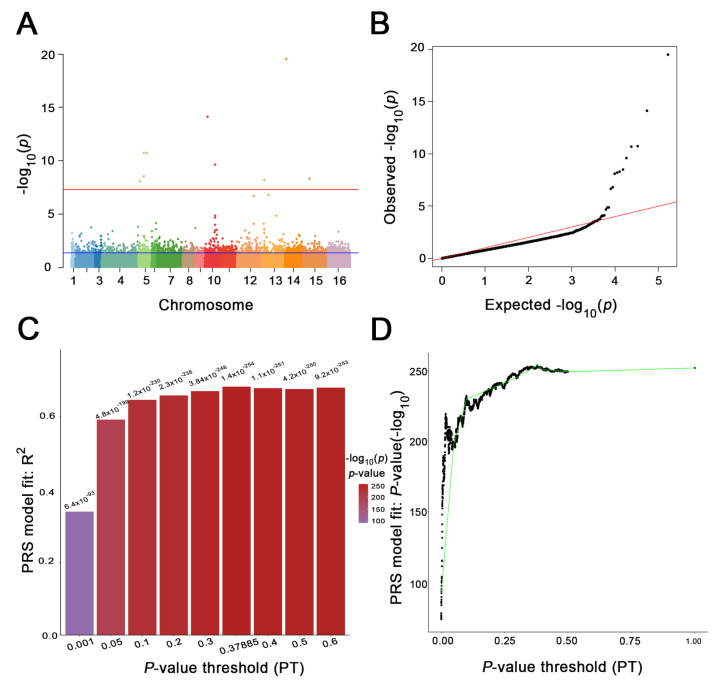
Genome-wide association study and polygenic risk score prediction of YPD40. (**A**) the Manhattan plot of GWAS result. The red line indicates that *p*-value is 5 × 10^−8^, and the threshold is statistically significant; the blue line indicates that *p*-value is 0.37885, and the threshold is the precise *p*-value threshold (*p*-value = 0.37885). (**B**) Quantile–quantile plot showing good normality. (**C**) Bar plot from *PRSice-2* showing results at broad *p*-value thresholds for PRS predicting YPD40. A bar for the best-fit PRS from the high-resolution run is also included. (**D**) High-resolution *PRSice-2* plot for PRS predictingYPD40. The thick line connects points at the broad *p*-value thresholds.

**Figure 3 jof-08-00914-f003:**
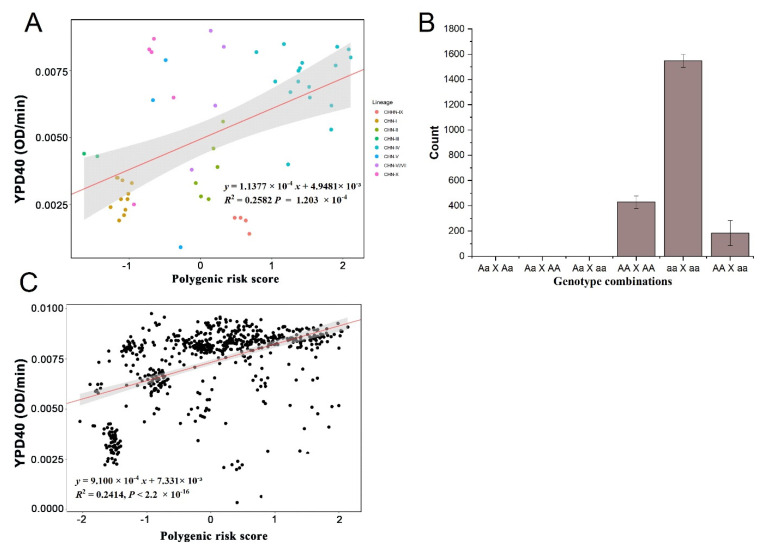
Relation between PRS and YPD40. (**A**) The linear relationship between PRS and 52 parental strains. (**B**) Distribution of combinations of genotypes on one locus from two parents of 613 F1 hybrids. The situations in which both parents were of homozygous genotype (AA×AA, aa×AA, aa×aa) accounted for 99.96% ± 0.04%. When both parents were homozygous genotypes, the genotypes of the F1 hybrids at this locus could be predicted with high accuracy. (**C**) The linear relationship between PRS and 613 F1 hybrids.

**Figure 4 jof-08-00914-f004:**
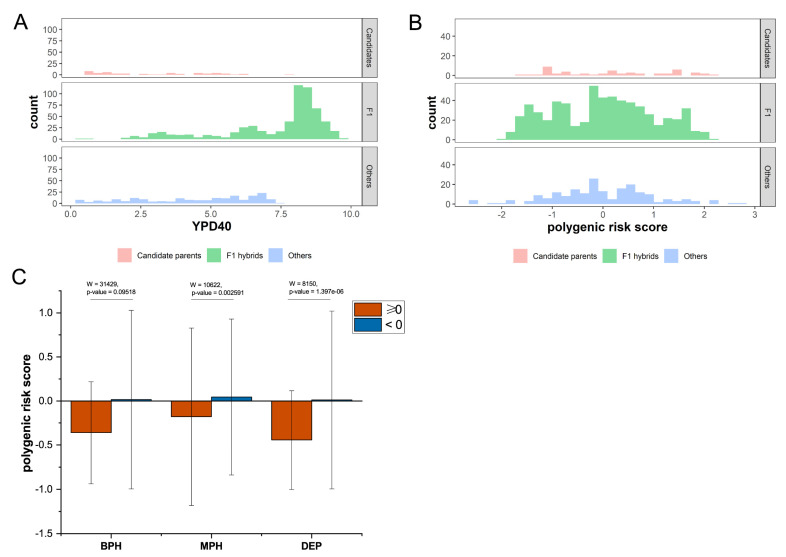
Distribution of YPD40 and PRS of F1 hybrids and 266 isolates for testing set. (**A**) Distribution of YPD40 of 52 parent strains and 613 F1 hybrids, as well as other strains in the testing set. The phenotype of F1 hybrids significantly exceeded that of parents and other strains, suggesting significant heterosis. (**B**) PRS distribution of 52 parent strains and 613 F1 strains, as well as other strains in the testing set. The parental strain and F1 strain had similar distribution range of PRS. (**C**) Relationship between heterosis of hybrid strains and PRS. BPH: best-parent heterosis, MPH: mid-parent heterosis, DEP: depression.

## Data Availability

Published data were used in this study.

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
