# Peer review of "Using Polygenic Risk Scores Related to Complex Traits to Predict Production Performance in Cross-Breeding of Yeast"

_jof, 2022, doi:10.3390/jof8090914_

Round 1
Reviewer 1 Report
The paper “Using Polygenic Risk Scores about Complex Traits to Predict Production Performance in Crossbreeding of Yeast” is focused on the development of a method for predicting the growth ratio of hybrid (S. cerevisiae) by identifying the polygenic risk score (PRS) based on genome-wide genetic association studies. The method proposed is interesting because is based on the predictable selection of individual from hybrid. However, if actually applicable at the level of starter strains, reduce the workload and improve work efficiency. From the application point of view, it is a method that requires great experience and specific databases. In the yeast starter sector, such as S. cerevisiae strains, the parameters of technological interest are many and this method would be difficult to apply. From the technological point of view it would also be useful to better understand the reproducibility of the results. The experimental plan is clear, but more details about the experiment in the methods section would make the potential application of the method more understandable in respect to the predictable results. Correct Saccharomyces cerevisiae in the text.
Reviewer 2 Report
Conclusions are not presented in manuscript, need to introduce.
There are some typo errors such as:
line 25 crossbreeding replace with cross-breeding
line 76 F1hybrids replace with F1 hybrids
line 164 each first of sentence need to be capitalized
line 237 crossbreeding instead cross-breeding
After the revision will be made it I recommend the publication of the manuscript.
Reviewer 3 Report
The MS "Using Polygenic Risk Scores about Complex Traits to Predict 2 Production Performance in Crossbreeding of Yeast" is well-written and of interest. There are only a few minor points to address.
Before moving forward the authors will have to "clean up" their manuscript. Please make sure you write the name of the organisms correctly (all in italics, genus name with capital, specie name no capital). See lns: 12 & 29.
Ln 91: insert the word "from" between 'origins' and 'around'
Ln 99: please rewrite, the meaning of the sentence is not clear.
Ln 102: write "wild-type" instead of 'wild'
Lns 104-106: rewrite sentence as "The genome sequence of S. cerevisiae S288c was used as the reference genome (version R64-1-1) was downloaded from the National Center for Biotechnology Information (NCBI)."
Ln 108: fix citation
Ln 113: fix citation
Ln 210: report p-value for non-significant finding
